# Vitamin D Deficiency in Older Patients—Problems of Sarcopenia, Drug Interactions, Management in Deficiency

**DOI:** 10.3390/nu13041247

**Published:** 2021-04-10

**Authors:** Małgorzata Kupisz-Urbańska, Paweł Płudowski, Ewa Marcinowska-Suchowierska

**Affiliations:** 1Medical Centre for Postgraduate Education, 01-813 Warsaw, Poland; emarcinowska1@gmail.com; 2Department of Biochemistry, Radioimmunology and Experimental Medicine, The Children’s Memorial Health Institute, 04-730 Warsaw, Poland; p.pludowski@czd.pl

**Keywords:** vitamin D, elderly, deficiency, sarcopenia, drug interaction, supplementation, treatment

## Abstract

Vitamin D deficiency frequently occurs in older people, especially in individuals with comorbidity and polypharmacotherapy. In this group, low vitamin D plasma concentration is related to osteoporosis, osteomalacia, sarcopenia and myalgia. Vitamin D levels in humans is an effect of the joint interaction of all vitamin D metabolic pathways. Therefore, all factors interfering with individual metabolic stages may affect 25-hydroxyvitamin D plasma concentration. The known factors affecting vitamin D metabolism interfere with cytochrome CYP3A4 activity. There is another group of factors that impairs intestinal vitamin D absorption. The phenomenon of drugs and vitamin D interactions is observed first and foremost in patients with comorbidity. This is a typical situation, where the absence of “hard evidence” is not synonymous with the possible lack of adverse effects. Osteoporosis and sarcopenia (generalized and progressive decrease of skeletal muscle mass and strength) are some of the musculoskeletal consequences of hypovitaminosis D. These consequences are related to an increased risk of adverse outcomes, including bone fractures, physical disabilities, and a lower quality of life. This can lead not only to an increased risk of falls and fractures but is also one of the main causes of frailty syndrome in the aging population. Generally, Vitamin D plasma concentration is significantly lower in subjects with osteoporosis and muscle deterioration. In some observational and uncontrolled treatment studies, vitamin D supplementation resulted in a reduction of proximal myopathy and muscle pain. The most conclusive results were found in subjects with severe vitamin D deficiency and in patients avoiding large doses of vitamin D. However, the role of vitamin D in muscle pathologies is not clear and research has provided conflicting results. This is plausibly due to the heterogeneity of the subjects, vitamin D doses and environmental factors. This report presents data on some problems with vitamin D deficiency in the elderly population and the management of vitamin D deficiency D in successful or unsuccessful aging.

## 1. Introduction

Vitamin D status has been widely studied in recent years and vitamin D deficiency remains a significant public-health problem. Research findings show that vitamin D deficiency is a global health problem for people, regardless of the region, age, sex and ethnicity [1,2]. During the recent decade, many studies have focused on vitamin D status, skeletal and extra-skeletal actions, and the key role of vitamin D receptor (VDR) in contributing to pleiotropic mechanisms. Vitamin D deficiency has been shown to be tied to many diseases, including autoimmune diseases such as rheumatoid arthritis, multiple sclerosis and type 1 diabetes. A deficiency of vitamin D is also associated with cardiovascular diseases (for example strokes), infectious diseases (bacterial, viral and, fungal), type 2 diabetes and some types of cancers (colorectal, breast and prostate gland). Recent research found that a concentration of 25-hydroxyvitamin D has an influence on various diseases and mortality [3,4,5]. However, a wide range of studies present some conflicting data regarding the impact of administering vitamin D supplements to the elderly.

The geriatric population is characterized by specific characteristics, such as age (youngest-old ages 65–74 years, middle-old ages 75–84, and oldest-old over age 85), successful and unsuccessful aging, functional status, comorbidity, polypharmacy and the presence or absence of geriatric giants. With the development of medical technology and improvement in healthy lifestyles, the average lifespan of human beings is increasing. Therefore, vitamin D is purported to be one of the factors that strongly influences aging with fewer negative outcomes and functional decline. This is also referred to as healthy aging.

Moreover, the deficiency of vitamin D in older individuals interfering with changes of musculoskeletal system, is purported to be one of the crucial factors predicting the risk of negative outcomes, including high risk of hospitalization, institutionalization, falls and loss of independence. The vitamin D status is found by measuring the total plasma *25*-hydroxycholecalciferol–25(OH)D concentration. The advisable 25(OH)D concentration (necessary to ensure skeletal and extra-skeletal mechanism of vitamin D) for all ages of the population is 30 ng/mL. However, the exact cut off point in clinical practice for the oldest of the elderly continues to be a subject of debate taking into account, among others, advanced aging processes, higher percentage of sarcopenic patients, more frequent multimorbidity and the fact that most studies and randomized controlled trials (RCTs) are conducted in a younger geriatric population.

The main focus of our publication is to highlight the impact of vitamin D deficiency in the middle-olds and older-olds with the loss of muscle mass, strength and function, as well as its influence on and relation to drugs, and also the possibilities of vitamin D management in sarcopenic patients.

## 
2. Sarcopenia


Geriatric giants, also referred to as geriatric syndrome, are characterized by three main traits—their prevalence is higher in the geriatric population in comparison with younger adults (also in the oldest-old in comparison to the youngest-old or middle-old), the reasons for this are multifactorial, but foremost the consequences have a stronger impact on geriatric patients than in younger adults. Sarcopenia (loss of muscle mass, function and performance) has been described as a new geriatric giant strongly influencing unsuccessful aging [6]. Research focused on sarcopenia is growing quickly, but many areas of this issue are still under discussion. The role of skeletal muscle loss (quantitative and qualitative decline) seems to be well established. The importance of vitamin D in skeletal muscle metabolism has become prominent in recently published research [7].

### 2.1. Definition

For many years, the definition of sarcopenia recognized by researchers, as well as clinical practitioners, has remained unsettled. Therefore, among others, the Foundation for the National Institutes of Health (FNIH) attempted to fill a critical gap in the research concerning muscle decline: “What degree of low lean mass is clinically relevant, when it is empirically grounded to its relationship to strength and function?” Studenski et al. considered the presence of weak hand grip strength as an essential condition to be stated as a reason for weakness, proposing critical thresholds for lean mass and muscle strength and function (a hand grip strength < 26 kg in men and <16 kg in women, with the grip strength adjusted for a Body Mass Index < 1.0 in men and <0.56 in women); assess appendicular lean mass (ALM), an ALM adjusted for BMI < 0.789 in men and <0.512 in women (or, ALM < 19.75 kg in men and <15.02 kg in women) [8].

In 2018, the consensus of the European Working Group on Sarcopenia in Older People (EWGSOP 2) published a definition and diagnostic criteria for sarcopenia that aimed to promote not only studies on the issue, but also screening for sarcopenia and caring for patients in everyday practice. It was also a revision of the established knowledge (including first consensus of EWGSOP published in 2010) on sarcopenia, its criteria and diagnosis. Authors of the publication underscored the role of the evaluation of muscle strength and physical performance, taking into account low muscle quality and quantity. For the first time, the cut-off points were emphasized for all measures in the definition of sarcopenia—for low muscle strength, hand grip strength < 27 kg in men (<16 kg in women) and chair stand > 15 s for 5 rises; the cut-off points for low muscle quantity being an ASM—appendicular skeletal muscle mass < 20 kg in men (<15 kg in women) and an ASM/height^2^ < 7.0 Kg/m^2^ in males (<5.5 Kg/m2 in females) and for low physical performance: a gait speed of ≤ 0.8 m/s, an SPPB (Short Physical Performance Battery) of ≤8 point score, a TUGT (Timed Get Up and Go Test) of ≥ 20 s and a 400-m walk test not completed or ≥6 min [9].

### 2.2. Outcomes

Mijnarends and colleagues emphasized that in older, institutionalized individuals, lower gait speed and chair stand (measures of low physical performance and decreased muscle strength) were potential factors of impairment in activities of daily living (ADL) and that such impairment was associated with lower quality of life in addition to higher healthcare costs [10].

Sarcopenia has a strong impact on the outcomes of the risk of falls and osteoporotic fractures, lack of independence and the inability to perform daily living skills, and is strictly associated with frailty phenomena. The frailty syndrome, a clinical characteristic (defined by Linda Fried as at least three traits consisting of reduced muscle strength, presence of fatigue or the individual’s reports of becoming easily fatigued, unintentional weight loss, a reduction in gait speed and a decreased amount of physical activity), remains to be a crucial factor contributing to unfavorable health outcomes in older adults, especially in the older-olds.

Moreover, muscle decline has been found to predict survival in elderly adults, and muscle strength seems to be a greater predictor of negative consequences than muscle mass [11,12,13]. It is believed that the statement of low muscle mass and decreased muscle strength or functioning could characterize a population of older adults at high risk of unfavorable health outcomes and could become a screening method, especially for the oldest-old [14].

### 2.3. Contributing Factors

It is also worth mentioning that sarcopenia, being a progressive process and a new geriatric giant, has many contributing factors—not only the aging processes (including hormone dysregulation: sex hormones, GH, IGF1, TSH, and insulin resistance; motor neuron loss; mitochondrial dysfunction), but also vitamin D deficiency, diet (including malnutrition or undernutrition), sedentary lifestyle (physical inactivity or immobility), diseases, drug treatments (secondary sarcopenia) and drug interactions.

## 3. Vitamin D Deficiency and Sarcopenia

The research regarding vitamin D deficiency attributed to different causes has shown that besides the osteomalacia, patients will frequently report muscular weakness, with incidences up to 97% [15]. In cross-sectional studies, as well as in prospective studies, low serum 25-hydroxyvitamin D concentrations were associated with increased risk of sarcopenia in elderly adults [16]. Moreover, the impact of vitamin D deficiency increases the reduction in age expression of skeletal muscle vitamin D receptor [17].

### 3.1. Interaction of Vitamin D with Skeletal Muscle

Several different mechanisms by which vitamin D interacts with skeletal muscle function have been found in cellular models. The first finding is that genomic impacts emerge from the interference of the 1,25-VDR-RXR (retinoid receptor) heterodimer at certain nuclear receptors that influence gene transcription. The second finding is described as non-genomic effects, distinguished by quick activation produced by other composite pathways of intracellular signal transduction subsequent to the binding of 1,25 (OH) D to its nonnuclear receptor [18,19,20].

The presence of the vitamin D receptor (VDR) in cytoplasm and nucleus facilitates rapid transcriptional actions. Down regulation of VDR during the lifespan has been assumed to be one of the factors contributing to muscle loss with age, and its deficiency interferes with muscle cell contractility [21].

### 3.2. Deficiency of Vitamin D as a Predictor of Sarcopenia and Frailty

The rate of functional decline and age-dependent atrophy of skeletal muscles in elderly individuals can be predicted by serum vitamin D levels [22]. There is a high prevalence of vitamin D deficiency in elderly populations, and VDR levels decrease in the muscles with advancing age [23]. Endo and colleagues underscored the role of VDR receptors with in vivo experiments. The size of muscle fibers was significantly smaller in VDR null mice than in wild mice. The loss of muscle mass happened faster in the study group (after 8 weeks of age) than in the control group, which suggests a key function for VDR receptors on muscle fibers, including a central functional and trophic role [24]. In addition, vitamin D is able to regulate the speed at which progenitors of skeletal muscle access the area of injury to repair and remodel [25].

Wilhelm-Leen and colleagues have observed that frailty and a low vitamin D status were associated in elderly men and women, with an overall 4-fold increase in the odd ratio of frailty [26].

### 3.3. Myostatin

It is widely known that vitamin D exerts its actions also by *other* mechanisms. Myostatin expression was found to be greater in the muscles of vitamin D-deficient rodents because vitamin D deficiency contributes to mitochondrial dysfunction and oxidative stress in muscle cells with decreases in superoxide dismutase (SOD) [27]. Furthermore, studies on human myocytes have highlighted the impact of vitamin D and VDR agonists on pathways known to regulate muscle aging (including ubiquitin ligases, the inflammatory biomarkers tumor necrosis factor-α (TNF-α) and interleukin 6 (IL-6)) [28,29].

### 
3.4. Vitamin D Deficiency and the Age-Related Skeletal Muscle Decline


Individuals with vitamin D deficiency have reduced muscle strength and a greater risk of falls, which is reversible with vitamin D supplementation. Also, in studies with animal models, (VDR knockout (VDRKO) mice) decreased grip strength, shorter steps, abnormal gait and decreased balance with shorter retention times signaling abnormal muscle coordination, have been observed. These defects in muscle function were observed to progress with advancing age as evidenced by a dose-dependent impact of VDR on grip strength [30,31].

Additionally, in elderly patients, long-term (12 months) nutritional intervention (cocktail including HMB-β-hydroxy-β-methylbutyrate, arginine, and lysine) only maintained muscle strength in older adults without vitamin D deficiency (>30 ng/mL), though the increase in the lean body mass was independent of the vitamin D status [32].

## 4. Vitamin D Deficiency and Osteosarcopenia

The possible influence of vitamin D deficiency and sarcopenia on bone mineral density (BMD) remains an issue of debate, even though there is a high prevalence of BMD in the elderly population throughout the world. Lee et al. investigated the interaction of vitamin D deficiency and sarcopenia with BMD in elderly individuals in Korea. They assumed that vitamin D deficiency and low BMD was more common in older patients with sarcopenia [33].

### 4.1. Definition

Osteosarcopenia—defined as the age-related concomitant loss of BMD, muscle mass and strength function—is a strong indicator not only of functional impairment, but also falls and fractures.

### 4.2. Risk Factors

The major risk factors for osteosarcopenia coincide with sarcopenia and are comprised of low physical activity and poor nutrition (taking into account especially low intake of protein, vitamin D and calcium, malnutrition and undernutrition). Sedentary lifestyle results in a loss of BMD, and muscle atrophy, due to reduced stimulation of muscle fibers and a reduction of the mechanical factors that promote osteogenesis [34].

### 4.3. Intervention

Multifactorial impact on osteosarcopenia, including multimodal exercises (resistance training, weight-bearing and/or balance training), protein intake, calcium and vitamin D supplementation, improves the process of muscle and bone decline and increases muscle mass and function. The supplementation of Vitamin D and calcium has been reported to improve BMD, muscle strength and to reduce falls and fractures in deficient adults who reside in elderly care communities.

Atlihan et al. [35] concluded that progressive resistant training is a safe and efficacious way to improve several characteristics of osteosarcopenia. Their research found improvement in total hip and lumbar spine BMD, muscle mass, quality and strength, but not in bone turnover or physical performance. However, the authors underscored that there were no RCTs identified that evaluated the effects of protein, vitamin D and calcium when compared to a control/placebo in osteosarcopenic subjects. Therefore, no current evidence supports non-pharmacological interventions, including, vitamin D, calcium and protein, in osteosarcopenic individuals [35].

The role of the pharmacological treatment of osteoporosis is well established and in clinical practice the algorithm adjusting treatment for individual patients is well known, but not for individuals with osteosarcopenia. There exist some data for some osteoporotic treatment (for example denosumab) concerning its protective role in sarcopenic patients—in falls reduction (as observed in the FREEDOM study) and also improvement in muscle function (measured by gait speed) was observed [36,37]. From the other side, in clinical practice, during osteoporosis treatment, unfavorable effects of drug–drug interaction is observed. It exists between bisphosphonates and, for example, vitamin D—during the simultaneous intake of both substances, bisphosphonates absorption in the digestive system is reduced. Therefore, the impact of osteoporotic fractures still needs further, extended studies.

## 5. Drug Interactions

### Drug Interactions with Vitamin D

Vitamin D status in humans is an effect of the joint interaction of all vitamin D metabolic pathways. Therefore, all factors that interfere with individual metabolic stages may affect 25-hydroxyvitamin D concentration in the circulation. To date, there is little hard evidence that agents such as lipase inhibitors, statins, antimicrobials, antiepileptics and others affect [25(OH)D] concentration in blood serum.

The agents with the potential to influence vitamin D status can be roughly divided into drugs that effect vitamin D intestinal absorption and those that influence vitamin D metabolism [38].

Included in the first category, lipase inhibitors are widely used for obesity treatment. They decrease triglycerides hydrolysis in the gut, causing an incremental rise of excreted fat from the typical 5% up to 30%. This increases fat-soluble vitamin D loss in the feces, at the same time decreasing the vitamin D pool available for absorption in the small intestine. In the second category of drugs that influence vitamin D metabolism, statins are an important class and are widely used as very effective agents in both the primary and secondary prevention of cardiovascular diseases.

All statins function as inhibitors of a rate-limiting enzyme in synthesis of cholesterol, namely hydroxyl-methyl coenzyme A (HMG-CoA) reductase. This action brings statins close to vitamin D metabolism, at the same time, suggesting their uniform action and similar side effects. Nevertheless, numerous studies demonstrated that the statins–vitamin D interaction give diverse and pleiotropic results. This is evident from a meta-analysis that was “inconclusive on the effects of statins on vitamin D with conflicting directions from interventional and observational studies”. Although the fundamental mechanism of action is identical for all the statins, they differ in water solubility and are catabolized in different ways depending on the statin type, patient’s age and vitamin D status, nutritional conditions and insolation.

It is known that atorvastatin, simvastatin and lovastatin are predominately metabolized by CYP3A4, a multi-substrate cytochrome also involved in vitamin D metabolites catabolism. Cytochromes in the CYP3A category are also very important enzymes in the vitamin D catabolic pathways. Therefore, any interference with their activity may cause a disturbance in the vitamin D status of the patient. It is known that some statins may compete for the active centers of the CYP3A enzymes, slowing down the catabolism of the vitamin D metabolites. This results in the vitamin D status increasing, especially in patients who are supplemented with vitamin D [39].

Antiepileptics (AEDs)*,* such as carbamazepine, oxcarbazepine, phenobarbital, phenytoin, primidone and valproate, have all been associated with bone health problems in epileptic patients. Taking into account that some of them are prescribed as co-analgesics, this type of therapy is widely used in clinical practice for elderly patients. AEDs are known to induce the enzymes from the catabolic pathway of vitamin D. This action results in a specific sequence of events leading to an increased fracture risk, beginning with the induction of hepatic cytochromes and the accelerated degradation of the vitamin D metabolites. AEDs can result in decreased vitamin D status and decreased intestinal calcium absorption, which has a negative effect on the circulating calcium pool [40].

Other drugs. The mechanisms of action and observational data suggest that other factors might also interfere with the metabolism of vitamin D. This group is comprised of glucocorticoids, immunosuppressive (for example cyclosporine, tacrolimus) and chemotherapeutic agents and highly active antiretroviral agents, as well as histamine H2-receptor antagonists.

## 6. Drug Interaction of Vitamin D and Sarcopenia

Some drugs could also play a role by preventing the loss of mitochondria, improving the function of endothelium and muscle metabolism. From that point of view, sarcopenia may be a potential curative choice for angiotensin-converting enzyme (ACE) inhibitors that are widely used in the geriatric population [41].

ACE inhibitors (ACEI) pretend to be one of the drugs that probably could present synergistic mechanism with vitamin D effects. There is evidence that indicates that sarcopenia and heart failure share several pathways and could possibly be improved by a common treatment plan. In animal studies, vitamin D deficiency is related to hypertension via the renin angiotensin system (RAS). The activation of VDR downregulates the RAS activity and inhibits the renin synthesis. Blood pressure, serum renin, and Ang II levels were found to be higher in VDR-null mice, [42,43].

Patients with severe heart failure present multiple histological abnormalities in muscle fibers, also called “cardiac skeletal myopathy” [44]. However, Zhou et al. stated that walking distance or muscle strength in older participants in RCTs did not significantly improve from treatment with ACE-inhibitors [45,46].

In a review published in 2021, Ekiz and colleagues aimed to expand the knowledge of the relationship between RAS dysregulation and sarcopenia, and also to provide some updated data concerning possible sarcopenia therapy (such as ACEIs, vitamin D, and exercise). Renin angiotensin system activity results in harmful effects on both the neuromusculoskeletal as well as the cardiovascular system. As such, therapy focused on the inhibition of the classical RAS hyperactivity seems to be essential in the management of sarcopenia and several other age-related pathologies. For this reason, treatment based on ACEIs combined with vitamin D, exercise and a healthy diet could have not only positive effects on modulation of the RAS, but also both physical and cognitive functions. This multimodal effect of ACEls and vitamin D could be profitable, especially in the oldest-old, when cognitive dysfunction may become a main factor accelerating the functional decline. Moreover, the non-classical axis that is enhanced has positive, multidirectional effects on the anabolic processes of the skeletal muscle and the cerebral blood flow but also on the neuromotor control system [47].


Nonetheless, insufficient evidence is available to recommend pharmacological interventions based on ACEI treatment administered with vitamin D in treatment of sarcopenia.


## 7. Vitamin D Supplementation and Treatment in Elderly

Recommendations for vitamin D intake in asymptomatic healthy individuals and healthy individuals with a high risk of vitamin D deficiency (which was published as the Central European Recommendation; similar to the Endocrine Society in USA) were published in 2018. These guidelines recommend the use of vitamin D supplements to achieve and maintain the optimal target 25(OH)D concentration in range of 30–50 ng/mL (75–125 nmol/m) [48]. This consensus of experts also constitutes the background for vitamin D management in sarcopenic patients. According to the recommendations, vitamin D doses larger than the tolerable upper intake levels (ULS) to prevent deficiency of the vitamin should not be prescribed. The ULS for adults and seniors with normal body weight is 4000 IU/d, but in obese adults and seniors, it is higher (10,000 IU/d) [49]. It is very important that treatment of vitamin D deficiency is based on 25(OH)D concentration and antecedent prophylactic management. Individual patients that have clinical risk factors for vitamin D deficiency (including medication with antiepileptic drugs and others which disturbing metabolism of vitamin D) and patients with bone diseases (fragility fractures, documented osteoporosis or high fracture risk, treated with antiresorptive medication, osteomalacia), should be treated.


The oral route (intake) of treatment is recommended (vitamin D2 and vitamin D3) and should be taken with food to aid absorption. The dosage should be adjusted according to the baseline deficit and the patient’s weight. The control level of 25(OH)D should be attained during treatment at the beginning and after 7–10 weeks.



Treatment of vitamin D deficiency should consist of 2 parts—the initial repletion phase of therapy (loading phase), and after the loading phase, initiating the maintenance.



The loading phase with vitamin D requires 7 to 10 weeks. The aim is to saturate all body compartments so the level of 25(OH)D is above 30 ng/mL (75 nnom/L) as it is indispensable for vitamin D extra-skeletal actions. During this time, loading doses of vitamin D (about 300,000 IU) should be given as daily split (divide) doses or intermittent doses every week. Single mega doses (300,000 IU to treat deficiency) are not recommended in the treatment of vitamin D deficiency. Maintenance regiments may be considered after the loading doses.



The rapid correction of vitamin D deficiency may be necessary in osteosarcopenic patients who are beginning treatment with a potent antiresorptive (such as zolenndronate, denosumab) or anabolic (such teryparatyd, abaloparatyd, romozosumab) agent. In these cases, the recommended treatment is contingent on split-loading doses (no single large doses) followed by regular maintenance therapy. Regarding the differences between cholecalciferol and calcifediol, including faster intestinal absorption of the calcifediol and linear increment uninfluenced by baseline vitamin D level, in elderly patients, often this type of therapy should be considered.


## 8. Vitamin D Management in Sarcopenic Patients

Remelli and colleagues concluded that the role of vitamin D supplementation in individuals with sarcopenia remains controversial. Several studies have explored the effect of oral vitamin D supplementation for the prevention of sarcopenia and frailty, yielding conflicting results [50]. In 2020, Nasimi and colleagues published the results of a double-blind randomized control trial conducted with sarcopenic older adults (a total of 66 older adults from 66 to 75 years). Their intervention was focused on fortified food (study group received yogurt fortified with 3 g HMB, 1000 IU vitamin D, and 500 mg vitamin C). After 12 weeks of nutritional intervention the research showed that consumption of fortified yogurt was associated with an increase in handgrip strength [mean change 4.36 (3.35–5.37) vs. 0.97 (−0.04 to 1.99)] and gait speed [0.10 (0.07–0.13) vs. 0.01 (0.00–0.04)] in the intervention group when compared to the control group (*p* < 0.001). In addition, the results showed an appreciable increase in vitamin D and IGF-1 levels in the intervention group (*p* < 0.001) [51]. However, the conducted studies differ strongly in the amount of vitamin D doses, period of time, tests used for muscle strength and function evaluation and they are not comparable for the population of the oldest-olds.

Cruz-Jentoft et al. emphasized the benefits of vitamin D supplementation in preserving muscle mass, strength, and physical function in elderly patients; also, in the prevention and treatment of sarcopenia, when included in multicomponent oral nutritional supplements. Among others, the authors underscored improvement in muscle mass and lower limb function (after 13 weeks) in the RCT named the PROVIDE (intervention consisted of vitamin D and leucine-enriched diets in older adults with sarcopenia). Another RCT conducted by Bo et al. found that muscle mass [relative skeletal mass index (RSMI), muscle strength and anabolic markers, such as IGF-I and IL-2, in older adults with sarcopenia can be significantly improved with combined supplementation with protein, vitamin D and vitamin E [52].

It has been underlined that the characteristics of patients should take into account not only the range of deficits but also the level of expected compliance [53].

## 9. Effect of Vitamin D, Protein Supplementation on Sarcopenia

### Vitamin D, Protein

In a meta-analysis, Gkekas and colleagues studied the effect of vitamin D and protein supplementation on sarcopenia, taking into account the EWGSOP 2010, AWGS 2014 or EWGSOP 2019 international criteria. It is worth mentioning that there were no data concerning the effect of monotherapy based only on vitamin D. Additionally, the authors stress some limitations of their study. Primarily synergistic end result of exercises and protein supplementation cannot be excluded. Furthermore, optimal vitamin D dosage and duration is still subject to discussion and there is lack of clear evidence in this field. Moreover, there were insufficient data to define the contributory role of baseline 25(OH)D levels; also the number as well as the sample sizes of the studies were relatively small. However, they assumed that, taking into account a total number of 776 patients from studies that met the eligibility criteria for qualitative and quantitative analysis, vitamin D (100–1600 IU/day) plus protein (10–44 g/day) supplementation had a positive effect on muscle strength, as demonstrated by improved handgrip strength and a decrease in chair stand time when compared with a placebo. Nevertheless, the effect on muscle mass, estimated by skeletal muscle index, was insignificant and no effect on muscle performance (estimated by walking speed) was observed with vitamin D plus protein [54].

It is probable that the effect of vitamin D on sarcopenia may be stronger in patients with low 25(OH)D concentrations. Notably, one study reported that muscle strength improvement in patients with 25(OH)D concentrations may be diminished in vitamin D-deficient populations resulting in decreased muscle strength [55].

In 2019, the International Conference of Frailty and Sarcopenia Research (ICFSR) formulated recommendations for frail individuals. The authors did not recommend the systematic supplementation of vitamin D for the treatment of frailty unless vitamin D deficiency is present. However, the recommendation of vitamin D supplementation was formulated as Consensus Based Recommendations (CBR) and the authors pointed out that further study is needed on the role of vitamin D supplementation in frailty prevention and treatment [56].

Also, the data on the influence of vitamin D supplementation on falls among frail older adults are inconclusive. The reduction of falls was observed in subjects with daily doses of 700 to 1000 IU vitamin D, contrary large bolus applications (monthly 60,000 IU to 100,000 IU of vitamin D or annual dosing of 300,000 IU to 500,000 IU) increased fall risk among frail older adults [57,58,59,60,61].

It should be also taken into account that the U-shape relationship between circulating 25(OH)D levels and frailty was reported only in women. It is likely that this phenomenon could be explained, among others, as a result of higher vitamin D binding protein (DBP) in females than in males. Consequently, total 25(OH)D levels can be higher in females comparing to males [62,63,64]. The role of DBP in sarcopenic patients should be also taken into account, but as highlighted by Bouillon et al., some of the major remaining research problems in this field concern the validation of measurements (for DBP, free 25(OH)D and free 1,25 1,25(OH)_2_D) and the extended knowledge about clinical impact of DBP gen polymorphism as well as impact on inflammation [65].

## 10. Conclusions

The last decade has provided us with an extensive body of knowledge concerning vitamin D in older adults. However, there remains a lack of sufficiently reliable and comparable data from long term RCT studies in oldest-olds that could deepen our knowledge in the field of pleiotropic mechanisms of vitamin D in geriatric patients with multimorbidity and polypharmacy in sarcopenia and osteosarcopenia treatment. As highlighted by Gkekas et al., the most efficacious approach at this time for clinicians who treat patients with sarcopenia is to combine physical exercise and nutritional supplementation into an individualized treatment plan. Taking into account recent studies, especially for the oldest-old who are the group of high risk of sarcopenia negative outcomes, multifactorial intervention should be recommended (including vitamin D management, protein intake and physical exercises). Moreover, further directions of studies in subgroups (youngest-old, middle-old and oldest-old) are needed to highlight the role of aging processes and the influence of the above-mentioned factors, as well as the target 25(OH) level for supplementation in this group. In 2021, the working group of experts has published consensus recommendations focused on the standardization of designs and outcomes, taking into account specific aspects of trial design. It will not only expand our knowledge but will also let us compare subjects and pro-myogenic therapies across studies [66].

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
