# Peer review of "Vitamin D Deficiency in Older Patients—Problems of Sarcopenia, Drug Interactions, Management in Deficiency"

_nutrients, 2021, doi:10.3390/nu13041247_

Round 1
Reviewer 1 Report
Specific comments:
Reference #: please change from I,II..XI to # 1, 2..
- Introduction
Low serum 25-hydroxyvitamin D concentrations were associated with increased risk of sarcopenia in elderly adults. The author should define how to diagnosis of vitamin D deficiency in elder population. Still < 20 for vitamin D deficiency and <30ng/ml as insufficiency in elder patients with comorbidities? Please define: youngest-old, middle-old, and oldest-old. Does your review focus on middle to oldest old?
- Sarcopenia
2.3 The pathology behind sarcopenia in elderly might be multi-factors, nutrition (vitamin D), medication and less physical activities. You may need include race, gender and socio-economic impact on sarcopenia. It looks like the main data are from CC population.
- Vitamin D deficiency and sarcopenia:
- A central controversy in the field of vitamin D clinical research is how to define vitamin D deficiency in different healthy population especially in geriatric population (1). Low total 25(OH)D levels may reflect poor nutritional status with low albumin and low vitamin D binding protein (DBP) levels (2) while free 25OHD might remained relative stable.
- High prevalence of vitamin D deficiency or low total 25OHD levels in elder population with sarcopenia Any data about 25OHD levels of sarcopenia elder patients compared to age matched healthy elderly population?
- VDR levels expression low in sarcopenia elder patients, any date about vitamin D metabolites change? like 1,25OHD and 24,25OHD levels in this population.
- Vitamin D deficiency and osteosarcopenia
Any difference of Vitamin D deficiency between sarcopenia and osteosarcopenia? any management different? We should include vitamin D plus calcium (plus medication) for patients with osteosarcopenia.
- Drug interactions
a.Drugs with the potential to influence vitamin D status can be roughly divided into drugs that effect vitamin D intestinal absorption and those that influence vitamin D metabolism [xxxvi]. Some agents might also affect vitamin D binding protein (DBP) and low total 25OHD might ≠ vitamin D deficiency.
b.You may need to add effect of anti-Parkinson and osteoporosis medications.
- Drug interaction of vitamin D and sarcopenia
ACEI and ARB are medication for HTN and CHF which may effect on vitamin D metabolism. You may suggest treating HTN in elder sarcopenia with ACEI or ARB and not other anti-BP medications (beta or calcium channel –blockers)? We do not treat sarcopenia elder patient without HTN?
- Vitamin D supplementation and treatment in elderly
- You should focus on vitamin D supplementation on elderly with sarcopenia, anything special or recent literatures.
- line 310-312: “ There are two essential points about supplementation in the healthy population. First, measurements of [25(OH)D] should not be tested before and during supplementation” why not test before or during supplementation? We should test before t supplementation o target deficiency and during supplementation to monitor possible side effects.
8.Vitamin D deficiency treatment
- It is very important that treatment of vitamin D deficiency is based on [25(OH)D] concentration <20ng/ml and antecedent prophylactic management. However, Individual geriatric patients with chronic hepatic diseases, renal diseases, medication and others which disturbing metabolism of vitamin D and DBP (2) and cause low total 25OHD levels.
- you should focus on vitamin D deficiency treatment in elderly or elderly with sarcopenia.
- Vitamin D management in sarcopenic patients
- The meta-analysis in 2020 show the effects of vitamin D treatment in sarcopenic individuals.
- 25OHD level differences between elderly patient with and without sarcopenia.
- Effect of vitamin D, protein supplementation on sarcopenia
Do you recommend vitamin D plus protein better than vitamin D alone?
Again, how to define vitamin D deficiency in these patients? (<20 or 30ng/ml?)
- Conclusion
You should have some conclusion: vitamin D deficiency plays a crucial role in developing sarcopenia? And management should be combination approach including vitamin D supplement, protein and physical therapy? any further directions in this field?
- P Youselzadeh et al Int J Endocrinol, vol. 2014; 2014. doi:10.1155/2014/981581
- N Jassil, et al. Endocr Pract. 2017;23: 605-61
Author Response
Dear Sirs:
Thank you for your suggestions and detailed comments. Below, are my answers to your remarks and suggestions.
Reviewer #1:
Form of references has been changed.
Ad 1. Youngest-old, middle-old, and oldest-old were defined and the main focus was underlined in the manuscript, as well as vitamin D deficiency definition.
Ad.2 The multifactorial pathology of sarcopenia has been underscored in the manuscript in the part concerning sarcopenia - contributing factors. Data concerning sarcopenia throughout the world show the crucial role of aging processes in muscle loss mechanisms. That’s why the authors decided to focus on aging processes taking into account that among women or subjects with low socio-economic status prevalence of sarcopenia can be higher, but aging remains to be the most influential factor.
Ad.3-4, 1-2 – vitamin D deficiency and sarcopenia
Thank you for this comment. The cut-off points of vitamin D in the oldest-old is still an issue of debate, because of poorer qualitative and quantitative nutritional status of geriatric population, low vitamin D binding protein levels and vitamin D receptor expression. However, based on our best knowledge in clinical practice, we use recommended 25(OH)D plasma concentration as a measurement of vitamin D status.
In VDR knockout mice (VDRKO) decreased grip strength, shorter steps, abnormal gait were observed. However, there is a lack of well-established data in humans concerning the influence of lower VDR expression on 1,25(OH)D or 24,25(OH)D levels or RCTs comparing the vitamin D status between sarcopenic and non-sarcopenic patients. This is remains an area of research that should be conducted.
Ad. 3 Vitamin D deficiency and osteosarcopenia
Undoubtedly, there is still a strong need to conduct a research concerning the epidemiology and treatment of osteosarcopenia. Many studies are focused only on sarcopenia or only on osteoporosis. Nevertheless, clinical observations show us that these two illnesses have similar pathology, but the exact epidemiology of the coexistence of them is currently unknown in oldest olds.
Vitamin D and calcium supplementation has been shown to improve BMD, muscle strength and reduce falls and fractures in deficient community-dwelling adults, however that there were no RCTs identified for multifactorial intervention.
The role of osteoporosis treatment was added to this section.
Ad. 5 Drug interaction
As stated in the introduction, the current measurement of 25(OH)D concentration is well established method in human beings determine vitamin D status and the exact role of agents such as vitamin D binding protein needs further studies. For that reason, we decided focus on 25(OH)D concentration.
Osteoporosis treatment was added in the section concerning osteosarcopenia.
Vitamin D deficiency is evident in Parkinson disease, and this deficiency significantly affects both motor and cognitive symptoms. There is evidence of abnormalities in the vitamin D-endocrine system in PD patients, including low bone mineral density (BMD), decreased vitamin D levels, and increased bone turnover makers. In PD it is the interaction of illness – vitamin D that plays the crucial role.
Ad. 6 Drug interaction and sarcopenia
As it was pointed in this section, there is insufficient evidence to recommend pharmacological interventions based on ACEI treatment administered with vitamin D in treatment of sarcopenia.
Ad. 7 /8 Vitamin D supplementation and vitamin D treatment
For individuals with sarcopenia, the recommendations for vitamin D supplementation and treatment are the same as with the general geriatric population. It is not recommended to test before or during supplementation of healthy individuals. Therefore, sarcopenic patients are often treated with vitamin D as they are symptomatic. As it was pointed out in the manuscript, treated patients are tested for vitamin D concentration. However, the main goal of vitamin D management is to reduce the deficiency in the entire elderly population as it constitutes the basis for many pathologies.
In 2019 International Conference of Frailty and Sarcopenia Research (ICFSR) formulated recommendations for frail individuals (including sarcopenia as a main characteristic of FS). The authors did not recommend the systematic supplementation of vitamin D for the treatment of frailty unless vitamin D deficiency is present.
As it was pointed above, and added in introduction section deficiency defined as 25(OH)D < 30ng/ml.
The section concerning meta-analysis was adjusted.
In the literature there is lack of population data comparing the oldest old with/without sarcopenia. Some data concerning sex –specific differences and DBP were added.
Ad. 11 Conclusion
An underscored conclusion concerning also recommendation of vitamin D plus protein intake was added to the manuscript as well as the conclusion concerning the necessity and direction of future studies

Reviewer 2 Report
This is a clear and concise review on the issue of Vitamin D deficiency in the elderly population, especially in sarcopenia, and its management considering also drug interaction.
However I would suggest to address also the issue of sex-specific differences in Vitamin D deficiency and VDR expression/function and in factors associated with aging and lifespan, such as sarcopenia and disease development which are increasingly recognized.
Minor typos:
Abstract:
Lines 18-19: Osteoporosis, sarcopenia ARE
Line 23 substitute "participants" with SUBJECTS or PATIENTS
Introduction:
lines 56: "substitute interfering with" with AND
3. Vitamin D deficiency and sarcopenia
line 130: substitute have con HAS
Author Response
Reviewer 2
Thank you for your remarks and comments.
Some data concerning sex –specific differences, the necessity and direction of future studies were added. The role of VDR receptor was commented in Interaction of vitamin D with skeletal muscle section.
Suggested phrases in the manuscript were changed.

Round 2
Reviewer 1 Report
Almost all societies recommended that < 20 for vitamin D deficiency and <30ng/ml as insufficiency in general population.
However, the exact cut off point for the oldest of the elderly remains still an issue of debate: Thus low total 25OHD might no might not equal to vitamin D deficiency in middle-old or old-old population ( ).
- Vitamin D deficiency and sarcopenia:
A central controversy in the field of vitamin D clinical research is how to define vitamin D deficiency in different healthy population especially in geriatric population (1). Aging and races might affect on DBP and albumin levels and then change total 25OHD levels (1,2).
- High prevalence of vitamin D deficiency or low total 25OHD levels in elder population with sarcopenia: Any data about 25OHD levels of sarcopenia elder patients compared to age matched healthy elderly population?
However, based on our best knowledge in clinical practice, we use recommended 25(OH)D plasma concentration as a measurement of vitamin D status. Aging with sarcopenia might have low albumin and DBP levels and lower total 25OHD levels .
Vitamin D supplementation and treatment in elderly
Do you have a target 25(OH) D level (as 25(OH)D ≥ 30ng/ml in elderly during supplementation to monitor possible side effects.
8.Vitamin D deficiency treatment
Again you should focus on vitamin D deficiency treatment in elderly or elderly with sarcopenia. This section should be deleted.
- Vitamin D management in sarcopenic patients
As you suggested that for individuals with sarcopenia, the recommendations for vitamin D supplementation and treatment are the same as with the general geriatric population. Why you need 3 different sections? You might need to combine section 7,8 and 9 into one or 2 sections.
- Effect of vitamin D, protein supplementation on sarcopenia
Do you recommend vitamin D plus protein better than vitamin D alone?
- Conclusion An underscored conclusion concerning also recommendation of vitamin D plus protein intake was added to the manuscript as well as the conclusion concerning the necessity and direction of future studies
further directions, not further studies in this field. also conclusion should be concise.
Do you think define vitamin D insufficiency or deficiency and the target 25(OH) level for supplementation are the possible further direction in this area?
- P Youselzadeh et al . Int J Endocrinol, vol. 2014; 2014. doi:10.1155/2014/981581; 2.N Jassil, et al. Endocr Pract. 2017;23: 605-61
Author Response
Dear Sirs:
Thank you for your suggestions and detailed comments. Below, are my answers to your remarks and suggestions.
Ad 1
In the manuscript, we focused not exactly on deficiency or insufficiency detected by blood tests but rather on the level needed to ensure skeletal and extra-skeletal mechanism of vitamin D .Usually, in the oldest olds vitamin D deficiency constitutes only one of many existing problems that interfere with. As it is not only the vitamin D deficiency but also advanced aging processes, multimorbidity, different level of functional decline, socioecenomical status, different severity of sarcopenia etc. it is very difficult to create homogenic group of oldest olds and usually the small percentage of subjects is representing this population. Therefore an explanation was added in manuscript
Ad. 3 Vitamin D deficiency and sarcopenia:
In the longitudinal Newcastle 85+ Study, Granic et al. s found that the lowest vitamin D season-specific quartile was associated with a higher rate of muscle strength decline in men aged > 85, but (TUG) did not differ across vitamin D quartiles.
As highlighted Remelli et al. also some prospective studies have examined the vitamin D role in older adults but there is high heterogeneity (of participants characteristics, type of muscle strength assessment) and only a few of them included persons aged 85 and older. However, this is the subgroup wit higher risk of low vitamin D status, sarcopenia, and functional decline.
Probably it is very difficult to distinguish in older olds what pathomechanism has played the crucial role at the beginning of the process and there is lack of data comparing homogenic elderly subgroups sarcopenic/ healthy.
It seems probable that aging with sarcopenia might have low albumin and DBP levels and lower total 25OHD levels, but as one of the last review highlited whether free 25OHD is a better predictor than total 25OHD for health outcomes is controversial and there is lack of validation for measurements (also for DBP), undoubtedly research in this field are needed. This aspect was also added to the manuscript.
.
Ad 7,8, 9 Vitamin D supplementation and treatment in elderly
During supplementation the risk of side effects is negligible if vitamin D doses are not larger than the tolerable upper intake levels , as it was described in the manuscript; only single mega doses (300 000 IU to treat deficiency) are not recommended neither as supplementation nor as treatment
This sections were combined, same parts were delayed , however some parts of them for the background of management of vitamin D in sarcopenic elderly.
Ad 10 Effect of vitamin D, protein supplementation on sarcopenia
It seems that vitamin D plus protein better than vitamin D however there is lack of sufficient data concerning only vitamin D supplementation in sarcopenic patients and multifactorial intervention should be recommended as it was concluded in the manuscript.
Ad 11.Conclusion
Probably the target 25(OH) level for supplementation is one of the the possible further direction in this area it was also added and conclusion was shorted.re-phrased for being more concise.